# Surgical Principles of Primary Retroperitoneal Sarcoma in the Era of Personalized Treatment: A Review of the Frontline Extended Surgery

**DOI:** 10.3390/cancers14174091

**Published:** 2022-08-24

**Authors:** Paula Munoz, Pedro Bretcha-Boix, Vicente Artigas, José Manuel Asencio

**Affiliations:** 1Department of General Surgery, Hospital Quironsalud Torrevieja, 03184 Torrevieja, Spain; 2Department of General Surgery and Surgical Oncology, Hospital Quironsalud Torrevieja, 03184 Torrevieja, Spain; 3Department of Surgery, Faculty of Medicine, Universidad Autónoma de Barcelona, 08193 Barcelona, Spain; 4Department of General Surgery, CSUR Sarcoma and Mesenchymal Tumors, Hospital General Universitario Gregorio Marañón, 28007 Madrid, Spain

**Keywords:** retroperitoneal sarcoma, liposarcoma, leiomyosarcoma, solitary fibrous tumor, compartmental surgery

## Abstract

**Simple Summary:**

Surgery is the only curative treatment for localized disease in retroperitoneal sarcoma (RPS). Frontline extended surgery, or compartmental surgery, is a recent surgical strategy consisting of resecting the tumor together with adjacent organs, with the aim of minimizing marginality. This review provides a practical step by step description of this standardized procedure, tailored to histologic behavior, tumor localization, and patient condition.

**Abstract:**

Surgery is the key treatment in retroperitoneal sarcoma (RPS), as completeness of resection is the most important prognostic factor related to treatment. Compartmental surgery/frontline extended approach is based on soft-tissue sarcoma surgical principles, and involves resecting adjacent viscera to achieve a wide negative margin. This extended approach is associated with improved local control and survival. This surgery must be tailored to tumor histology, tumor localization, and patient performance status. We herein present a review of compartmental surgery principles, covering the oncological and technical basis, and describing the tailored approach to each tumor subtype and localization in the retroperitoneum.

## 1. Introduction

Retroperitoneal sarcomas (RPS) represent <1% of all adult malignancies, and account for about 15% of all soft-tissue sarcoma with s (STS). Under the definition of RPS we find a large number of different histologies, whose common characteristic is its critical anatomical location [1]. Arising from the retroperitoneal space entails a growth in a “compartment” without anatomical limits and surrounded by vital structures. Hence, there are no “low-risk histologies” as with STS of the trunk and extremities, and even low-grade RPS have high rates of local failure, which undermine long-term survival. The anatomical complexity of the retroperitoneum and the biological heterogeneity of the different RPS entail difficult challenges for their management. Surgery is the key element in RPS treatment, being the only curative therapy in localized disease. Retrospective historical series evidenced that R0 resection was the most important prognostic factor related to treatment, and the only factor where the surgeon could make a difference [1,2]. In recent years, there has been a progressive increase in R0 resection rates, up to 70–95%, which shows a trend towards more aggressive surgical policies and a better selection of patients [3,4]. With the aim of improving long-term oncological results of RPS, a new surgical strategy based on the principles of compartmental surgery for STS of extremities has been proposed. This surgical approach, known as frontline extended resection, consists of an en bloc resection of the tumor together with adjacent organs, with the aim of minimizing marginality (R1-2 resections) in the retroperitoneum. This surgical approach has been standardized and is currently the recommended approach by the main groups of experts [5,6,7,8]. However, in RPS there are multiple prognostic factors such as histologic subtype, tumoral grade, age, or multifocality which can predict the pattern of recurrence and may guide the aggressiveness of the surgery or the need for complementary systemic treatment [9]. In this article we will review the surgical and oncological principles of frontline extended resection, outlining the steps of this standardized procedure. We will provide the rationale for personalized treatment for RPS guided by histologic subtype, localization in the retroperitoneum, and patient characteristics.

## 2. Frontline Extended Surgery: Background and Bases

The compartmental surgery for STS of extremities is based on the resection of the anatomical compartment where the tumor lies and has demonstrated that resection margin status is key on disease-free survival and overall survival of sarcoma patients [10]. In STS, four types of resections are classically defined [11]: intralesional (R2), marginal (through the tumor pseudocapsule) (R1), wide (i.e., with a margin of healthy tissue), and radical (i.e., en bloc resection of the anatomical compartment). Since there are no defined anatomical compartments in the retroperitoneum, surgery for RPS will always be a marginal resection. Acknowledging the above, the need for an optimal resection strategy in RPS was postulated: the frontline extended approach, also known as compartmental surgery. This extended surgery includes the tumor and adjacent organs located at 1–2 cm from the tumor, including the colon anteriorly, kidney, and psoas muscle posteriorly, even if there is no macroscopic organ invasion. These organs can be safely and easily resected with limited impact or comorbidity, in contrast with other critical structures such as the duodenum, pancreatic head, or vertebrae, where a marginal resection is favored.

Through this approach, the ipsilateral retroperitoneal fat is resected by means of intrabdominal dissection beyond the anatomical barriers, ensuring the elimination of potential satellite metastases, optimizing surgical margins, and increasing the possibilities of obtaining an R0 resection. This also reduces the persistence of microscopic disease and tumor dissemination [4,11]. This extended approach can reduce local relapse (LR) rates, and, consequently, increase overall survival (OS) given that RPS local failure is the main cause of death related to the tumor, considering many patients will die without distant metastases. However, the appropriate extent of resection in RPS remains a topic of debate, while the benefit of converting R1 to R0 resection has not been proven, prevention of R2 resections is crucial. The impact of microscopic surgical margins in extremity and trunk sarcoma has been well established, which has not been the case with RPS. Nevertheless, surgery should aim to remove the tumor completely to minimize marginality. There are several reasons for this uncertainty on margin assessment in RPS. The main one is the lack of a standardized protocol for pathologic sampling, as RPS specimens are big masses, arising from a virtual space (retroperitoneum), which is difficult to replicate on the pathologist’s table.

A few years ago, A. Gronchi (Istituto dei Tumori, Milan) [5] and S. Bonvalot (Institut Curie, Paris) [6] first described the concept of compartmental surgery in the retroperitoneum, with the aim of improving local control of RPS. In 2009, these authors presented the first retrospective series, from a 20-year period, on oncological outcomes after the implementation of this new surgical policy on RPS treatment. The multicenter study by S. Bonvalot showed that simple resection was associated with a threefold higher rate of local recurrence than compartmental resection, while A. Gronchi and his group, analyzing the change in surgical policy in their institution (simple resection—before 2002—vs. compartmental resection 2002–2009), showed a decrease in local recurrence (LR) at 5 years of 48% vs. 28% in the extended surgery group. These studies showed that margin status was a key prognostic factor on LR as a consequence of a more liberal organ resection strategy. No significant difference in terms of completeness (R0-1 vs. R2) of resection was found between the two groups, nor could they prove an advantage in OS from compartmental surgery. However, subsequent follow-up of these series has shown that compartmental surgery can improve OS, especially in low- and intermediate-grade liposarcomas, with a morbidity and mortality comparable to that of other oncological abdominal surgeries. The study and effort of these two European groups consolidated the current scientific basis of frontline extended surgery for RPS [11,12].

The retroperitoneum cannot be considered a real compartment due to the absence of anatomical boundaries and the presence of vital structures at its limits. However, there are natural barriers to tumor dissemination, such as the fascia of the psoas muscle, the vascular adventitia, or the peritoneum, which help to define the surgical space. These anatomical barriers will be altered with large tumor growths or because of multiple reinterventions due to local recurrence. In RPS surgery, the encased organs such as the colon and mesocolon, the kidney, or the pancreas become the oncological margins of the RPS. The rationale for this extended surgery is the liberal resection of the ipsilateral colon, kidney, and adrenal gland—anterior margin—and psoas muscle—posterior margin—en bloc with the tumor, thus clearing all ipsilateral retroperitoneal fat. The vascular dissection is performed through the adventitial plane, and vascular resection of main vessels would only be performed in case of tumor invasion. Spleen, pancreatic tail, or diaphragm would only be removed with very large left upper quadrant tumors. More aggressive resections such as duodenojejunal junction, pancreatic head, rectum, bladder, or the vertebral bodies will only be performed in highly selected cases when a clear tumor infiltration exists.

Given the sacrifice of healthy organs without tumor invasion together with the tumoral mass, this multivisceral “liberal” resection policy has received considerable criticism [13,14,15], generating a heated debate within the scientific community about the oncological value of compartmental surgery in terms of OS compared to the morbidity and mortality associated with such aggressive surgical strategy. In order to evaluate the physiological reasons of this approach, some studies have analyzed the histopathologic organ invasion (HOI) of the resected organs together with the RPS. These showed that up to 25% of the resected organs without intraoperatively evidence of tumor invasion had HOI identified pathologically [14]. That fact justifies a more extended approach to secure margins in RPS. In 2017, a study from Dana Farber/Brigham and Women’s Cancer Center demonstrated that HOI was an independent predictor of adverse prognosis with a worse 5-year OS (34% vs. 62%, *p* = 0.04). In this study, 26% of resected organs demonstrated HOI [16]. However, HOI should be considered as a marker of biologic aggressiveness rather than the rationale for organ resection, which should be guided by RPS histology. The pathological evaluation of HOI is difficult and has not been standardized yet. In sarcomas, HOI should not be just defined as the infiltration of visceral parenchyma, but also as the tumor adherence to the organ. For instance, in RP-LPS, detaching the tumor from an adherent organ will guarantee an R1 resection. Regarding surgical morbidity, the major retrospective series reflects that these interventions are safe when carried out in reference centers counting with multidisciplinary teams (MDTs) with a high volume of patients, whereby pancreatic and vascular resections are associated with higher surgical risk [17]. It has been recently reported that there is not an association between surgical morbidity and long-term oncologic outcomes [18].

Recently, Callegaro et al. have published a multi-institutional study of 1942 RPS resected patients, investigating the outcomes related to the changes in treatment strategy during 2002–2017 at 10 sarcoma referral centers. This study describes how the rate of R2 resection decreased and the median number of resected organs increased over time. It concludes that the long-term survival of RPS patients who underwent resection had increased during the last 15 years, with the best survival outcomes at the last period of the study (2012–2017). This is a confirmatory study of how a better selection of patients and quality of surgery (i.e., a decrease in R2 rate and intraoperative tumor rupture, and the adoption of a more aggressive approach) increase disease-specific survival in RPS [19].

Due to the rarity and heterogeneity of RPS, we lack prospective randomized trials, and it is hard to extrapolate strong evidence from retrospective studies. However, an increase in survival rates in recent years in resected RPS denotes that a liberal en bloc resection of the tumor with adherent organs, even if not infiltrated, tailored to each histologic subtype, tumor localization, and patient performance status and comorbidities should be the gold standard for RPS surgery [20,21]. Compartmental surgery has been advocated by the masters of sarcoma surgery [9,22] and recommended by supporting guidelines [23].

## 3. International Collaboration in RPS Care

With the aim of optimizing the management of RPS universally, the Transatlantic Australasian Retroperitoneal Sarcoma Working Group (TARPSWG) was founded in 2013. This group is constituted by 150 tertiary centers of high volume sarcoma surgery and has the aim of integrating data of this rare disease. In recent years, the TARPSWG has generated a series of publications for clinical decision-making in RPS through guidelines and consensus papers on diagnostic and therapeutic actions, clinical guidelines, nomograms, etc. It also opened the opportunity to participate in multicenter prospective trials. The creation of collaborative groups contributes to progress on the management and understanding of this rare disease. Regarding RPS surgery, the current recommendation of the TARPSWG is to minimize the risk of microscopic positive margins through a standardized extended approach, especially at the first surgery [9,24,25].

## 4. Standardized Surgical Technique of the Frontline Extended Approach

In this section we describe the standardized surgical technique of the frontline extended surgery, as performed in the three subtypes of retroperitoneal liposarcoma (LPS) where it is the standard of care: well-differentiated LPS (WDLPS), G2 dedifferentiated LPS (DDLPS), and G3 DDLPS. For other histologies such as leiomyosarcoma, solitary fibrous tumor, or malignant peripheral nerve sheath tumor (MPNST), this approach varies depending on several characteristics, such as the distinction of the tumor boundaries, the recurrence pattern, and the risk of systemic failure. The targeted surgical approach guided by histology will be discussed in the text below. This approach has been previously described and standardized during the E-Surge and European Organization for Research and Treatment of Cancer (EORTC)–Soft Tissue and Bone Sarcoma Group (STBSG) meetings, and in several publications from the TARPSWG group [24,26].

The first step of the standardized technique of extended en bloc resection, is the planning of the incision and assessment of exposure. We advise the use of versatile abdominal retractors, such as the Thompson surgical retractor, with angled abdominal blades to obtain stable exposition of the surgical field and retraction of abdominal viscera.

The most common approach is a generous midline incision. It can be extended laterally in cases of bulky tumors, such as a flank or subcostal incision, to offer better exposure for vascular control. For RPS extending along the psoas and into the groin or leg, an oblique incision inferiorly may be needed (Figure 1). In tumors with retrohepatic or diaphragm extension, a thoracotomy is necessary to expose and control the diaphragm, the inferior cava vein (ICV) and its entry into the right atrium.

The recommended approach is the dissection from the medial aspect towards the lateral aspect. First, we evaluate a possible tumoral encasement of the bowel or its mesentery Figure 2A,B. If there are no unresectability criteria, we proceed with the surgery. The gastrocolic ligament, the transverse colon—distal limit of the colon division for right-sided tumors and proximal limit for left-sided tumors—, and, in right-sided tumors, the distal ileum are divided. In left-sided tumors, the distal division of the colon at recto-sigmoid junction is performed further on. If possible, the middle colic vessels are spared. The next step is the dissection of the medial margin by the exposition of the length of the IVC or the aorta towards the bifurcation of the ipsilateral iliac vessels. The vascular dissection must remove the adventitial layer with the RPS. During this step the renal vessels are divided. We recommend dividing the renal artery previously to the vein to avoid blood sequestration in the renal parenchyma to reduce the risk of hemorrhage (Figure 2C,D). Division of the inferior mesenteric vessels in left sided tumors is not mandatory but should be skeletonized, and a selective ligation of the sigmoid vessels is preferred to allow adequate vascularization of the rectum. The dissection reaches the ipsilateral gonadal vessels, which are tied. At this stage, the distal portion of the ipsilateral ureter must be identified and divided, along with the distal transection point of the sigmoid colon in left sided tumors (Figure 2E). Iliac vessels dissection must begin distally and progresses cranially, dividing all the vascular branches that nourish the tumor.

The surgery continues by approaching the posterior margin of the dissection. The prevertebral plane is reached, and the dissection of the iliopsoas muscle begins. The division of the medial fibers of psoas major muscle requires the identification, dissection, and preservation of the femoral and obturator nerves, located at its posterior and lateral edges. Preservation of the iliohypogastric, ilioinguinal, genitofemoral, and femorocutaneous nerves is further recommended, when possible (Figure 2F). We continue the resection of the fascia or the superficial portion of the psoas muscle from inferior to superior by detaching it from the lumbar spine and dividing it along L1-L4 vertebrae at its origin. The dissection ends with the resection of the ipsilateral parietal peritoneum from the lateral abdominal wall to the psoas fascia, continuing the peritonectomy up to the splenic or hepatic colon flexure (Figure 2G). The diaphragmatic peritoneum must be resected when it is abutted by the tumor, including the musculature if infiltrated, with its subsequent reconstruction.

Tumoral resection is then completed by removing the tumor en bloc with the ipsilateral kidney, the colon, and the anterior fibers of the psoas muscle along with its fascia, from the medial margin to the lateral margin. This approach allows safe vascular control in case of complex vascular involvement or accidental bleeding. The iliacus muscle and vertebral bodies become the posterior margin, the greater vessels of the abdomen and the intestinal mesentery the medial margin, and the abdominal wall muscles the anterior and lateral margin Figure 2H.

There are variations between right-sided vs. left-sided LPS, regarding the sequence of the surgical steps and the location of the tumor relative to the abdominal viscera (Figure 2 and Figure 3):Right side: right LPS may require a right medial visceral rotation en bloc with the tumor, to assess involvement of the IVC. A wide Kocher maneuver allows both duodenal retraction and access to the full extension of the infrahepatic cava vein. This approach will allow for the preservation of the duodenum and the head of the pancreas. The resection of the duodenum-pancreas does not improve disease-free survival and is in fact associated with the highest complication rates, therefore the aim of the surgery at this point is to pass through the tumor pseudocapsule (i.e., marginal dissection) releasing it from these organs. However, partial resection should be considered when the pancreatoduodenal junction dissection from the tumor leads to duodenal perforation due to a wall thinned by tumor compression or invasion. In extremely rare cases, duodenopancreatectomy is required.Left side: In left LPS a wide release of Treitz angle without injuring the duodenum is important. When there is a clear infiltration of the 3rd duodenal portion, we should add its resection with a reconstruction through a duodeno-jejunostomy. In left upper quadrant tumors, a distal spleno-pancreatectomy or even diaphragm resection is required [19,20].

## 5. Personalizing Surgery in Primary RPS

Relying on current and improved knowledge of the biology and behavior of RPS, it is advisable to pose a personalized treatment strategy. We have been able to better define the influence of different prognostic factors—age, tumor size, histological grade, and subtype, multifocality, and quality of surgery—in predicting the evolution and outcomes of each patient, thus guiding our multimodal therapeutic strategy [27]. That prognosis information has enabled the surgeon to do a more rationalized selection of those patients who will benefit from an aggressive surgical treatment versus those for whom it is advisable to avoid large resections, as well as establishing the need for a complementary systemic treatment.

Some prognostic nomograms have been developed in recent years [9,28,29] with the aim of predicting oncological outcomes, moving away from the classical TNM staging system, which has failed to be as reliable as it is in other solid tumors. These nomograms represent a predictive tool to guide the subsequent treatment in the postoperative setting, and are recommended by the AJCC 8th in primary RPS patients [30].

Until we have a preoperative prognostic tool, surgical strategy should be guided by the biologic behavior of each RPS and its recurrence pattern. Thus, three main factors should guide the sarcoma surgeon: histology, tumor localization, and patient-related characteristics.

### 5.1. Histology

Tumoral histology is a main prognosis factor that determines tumor aggressiveness and recurrence pattern, ultimately conditioning overall survival. Primary retroperitoneal sarcomas are a group of mesenchymal neoplasms arising at the retroperitoneum and are non-visceral in origin. Although there are more than 60 subtypes of sarcomas that can arise at the retroperitoneum, there are six most common histologies that account for 80% of primary RPS: well-differentiated LPS (WDLPS), dedifferentiated LPS (DDLPS), leiomyosarcoma (LMS), solitary fibrous tumor (SFT), malignant peripheral nerve sheath tumor (MPNST), and unclassified/undifferentiated pleomorphic sarcoma UPS [31] (Figure 4). Less common histologies such as synovial sarcoma or myxofibrosarcoma have also been described [32,33,34]. Benign entities such as retroperitoneal lipoma, schwannoma, and other malignant tumors such as gastrointestinal stromal tumors GISTs, desmoid type fibromatosis, visceral sarcomas, and adrenal tumors must be excluded from primary RPS classification [9,35,36].

Because different histologies result in different biological behaviors, the surgical approach should be tailored to each subtype and will be summarized in this review:

Frontline extended surgery offers the best advantage in low- and intermediate-grade tumors, where OS depends mainly on local control. In high-grade tumors and histologies with a great tendency to develop distant metastases (DM), the benefit of the extended surgical approach is called into question. In such cases, the aim of the treatment is an R0 surgery, although it is always complemented with an optimal systemic treatment to minimize the risk of distant metastases. This fact was well reflected in the largest published series of resected RPS, from A. Gronchi et al., 2016 and Callegaro et al., 2021 [4,19,36,37]. The authors highlighted the different patterns of recurrence of WDLPS and grade 2 DDLPS compared to grade 3 DDLPS grade 3 and LMS where DM were the main determinant of OS. Relying on these results, we know that LPSs are characterized by poorly defined dissection margins and a high prevalence of multifocality with adipose infiltration of the nearby organs; thus, the recommended surgery is an extended en bloc resection that includes adjacent organs requiring the ipsilateral retroperitoneal fat resection [14,26]. WDLPS is multifocal in up to 18% of tumors, consequently without clearing all ipsilateral retroperitoneal fat, there is a considerable risk of leaving tumor behind. Multifocality increases to 50% with local recurrence of retroperitoneal LPS. This strategy has shown to improve overall survival in these subtypes. WDLPS has a far better prognosis at 5 years than DDLPS, although the local failure rate is as high as ~39% at 5 years [11,26]. This local failure risk distinguishes retroperitoneal WDLPS from atypical lipomatous tumor (ALT) of the extremity, a low-risk entity which can be treated by marginal excision. This suggests differences in tumor biology between retroperitoneum and extremity LPS which are not apparent in histologic examination. Furthermore, it is precisely WDLPS and DDLPS grade 2 subtypes that benefit the most from frontline extended surgery [38].

Grade 3 DDLPS, additionally, has a high risk of systemic progression, which precludes the prognosis of these patients. In this group, a frontline extended approach should be complemented with a neoadjuvant or adjuvant systemic treatment to improve the overall control of the disease with a peri-operative agent which has action against micrometastases [39]. Of note in RPS, tumoral grade may be aligned with tumor biology and prognosis more than histology, as is the case with grade 3 DDLPS and grade 3 LMS, sharing similar survival outcomes [4,40,41]. In LMS where prognosis is hindered by systemic failure, the quality of the first surgery, consisting of an en bloc resection with vascular structures, is critical. In this group, the tumor’s borders are better defined and the limits with the neighboring organs are clearer, with a low rate of HOI when we can detach the tumor from adjacent structures. Thus, the adjacent organs not infiltrated by the tumor can be safely spared without compromising the radicality of the surgery.

SFT is usually a low-grade tumor, with low rates of LR and DM, and an indolent behavior. In most cases of retroperitoneal SFT, a resection of the tumor together with a rim of the adjacent fat will suffice. The aim of this resection is a negative margin and complete excision while minimizing morbidity. However, 10% of cases present with more aggressive tumors—larger than 10 cm, or with histological characteristics of malignancy—, in which case, compartmental surgery with resection of adjacent organs is recommended [42].

MPNST like LMS are high-risk tumors, where the aim of the surgery is to obtain R0 resection but preserving adjacent structures, which must be complemented with a multimodality treatment. These tumors are aggressive lesions requiring resection of the nerve at its origin resulting in unavoidable surgical morbidity. In any event, in such high-risk histologies, a complete R0 surgery is the treatment of choice and the only one curative, given the lack of other effective modalities. Table 1 summarizes the surgical management based on the biology and outcomes of the main RPS histologies. [11,19,35,41,43].

### 5.2. Localization

About 15% of all sarcomas arise at the retroperitoneum [45]. The retroperitoneum is a complex anatomic space located behind the peritoneal cavity and bordered by great muscles: the diaphragm, the abdominal wall, and paraspinous muscles. For the sarcoma surgeon a profound knowledge of the retroperitoneal anatomy and its relationship with major vessels and organs, the surgical dissection planes of this space, and the continuity with the extra-retroperitoneal spaces is crucial [44]. When we define RPS, we are including sarcomas located in the retroperitoneal space, but also those located in the pelvis, the psoas muscle, the inguinal canal, and the great vessels and nerves of the retroperitoneum. When the surgeon faces bulky tumors such as LPSs, this surgery should be planned and performed in a standardized manner, based on the laterality of the tumor, right or left, which will determine the anatomical approach, surgical steps, and the organs that will be resected.

Vascular or nerve tumors, such as primary IVC sarcoma or femoral nerve sarcomas, merit special consideration [43,46,47,48]. These tumors require a wide excision which will include a wide rim of the vessel or nerve from which they originate. These surgeries imply a greater morbidity. That is the case of the MPNST of the femoral nerve, where, due to the loss of function associated with resection, the patient should be made aware and informed of the impact of surgery [49]. In the case of IVC sarcomas, the assessment of resectability and the need for reconstruction will depend on the extent of the tumor, tumor localization relative to the renal veins [47,48], and the previous development of collateral circulation to restore the natural venous return. RPS with pelvic extension are usually adherent to the rectum and bladder peritoneum and they can extend throughout the sciatic or obturator notches, which must be studied preoperatively to plan the approach. Both the rectum and the bladder, as well as nervous structures, should only be resected if there exists tumoral infiltration. Given the high risk of positive microscopic margins at these localizations, the use of preoperative and/or intraoperative radiotherapy (IORT) is recommended [50,51,52].

When the tumor has an extension into the groin, we should continue the midline incision in an oblique direction downward along the leg to improve exposure and to avoid rupture of the tumor that would be fatal. Since resection of the inguinal ligament is usually necessary, reconstruction with a mesh or a pedicled myocutaneous flap should be planned preoperatively.

Psoas sarcomas—femoral nerve sarcomas, undifferentiated pleomorphic sarcoma (UPS), atypical lipomatous tumors, and schwannomas—should be approached as parietal sarcomas rather than RPS, as most of them arise in the muscle depth, thus the fascia will act as a natural barrier, and an extraperitoneal approach is sometimes possible [26]. Tumor histology and size will determine the surgical approach. In MPNST, whenever possible, we will perform an extraperitoneal approach to avoid the violation of peritoneal plane in anticipation of consecutive reinterventions. In the case of UPS, an aggressive tumor with a poor prognosis, a wide transperitoneal approach with midline incision with extension into the groin that allows adequate vascular control is usually required. For schwannoma or atypical lipomatous tumor, marginal resections can be considered through an extraperitoneal approach [49,53].

#### Localization and Technical Unresectability

Currently there is no consensus on what constitutes unresectability in primary RPS. Beyond anatomical limitations, we lack data regarding how locally advanced tumors can hinder disease-free survival after resection. Hence, current expert opinion bases unresectability as the inability to achieve complete macroscopic resection due to the involvement of major vascular structures—celiac trunk, superior mesenteric vessels, and aorta—, central mediastinal structures, and the spinal cord invasion [40]. Generally, it is assumed that in RPS without distant metastases, aggressive surgery is justified if R0 resection is possible. The need for extensive surgery, such as cephalic pancreatoduodenectomy (CPD) and right hepatectomy, is also considered a criterion for unresectability due to the higher risk of major complications [18]. The role of vascular resection and reconstruction outside the context of sarcomas of vascular or nerve origin is controversial, and should be limited when R0-1 resection is not possible, with a careful selection of patients. Current expert opinion supports resection of the IVC, the iliac vessels and a maximum of two suprahepatic veins. Some groups advocate vascular resection when there is tumoral infiltration to achieve R0 resection, as that may improve disease-free survival with an acceptable surgical risk, and an algorithm has been described for vascular resection and reconstruction strategy in RPS [46]. Nerve resection and reconstruction is also feasible, while requiring preoperative planning and a multidisciplinary approach supported by plastic surgery.

### 5.3. Patient

Like other oncologic surgeries, multimodal treatment in RPS must be individualized regarding patient-related factors such as age, comorbidity index, and performance status (PS). Patient-related variables to be weighed, in order to procure a common rationale for patient selection and treatment decision, have not been rigorously described or investigated.

Some previous reports have stated that, in elderly patients, more conservative surgery should be indicated, and that compartmental surgery could even be contraindicated, given that older age negatively affects prognosis in RPS [54]. However, an age limit has not yet been established, and the patient’s general condition should be the most determining factor in most cases [54,55,56]. Some comorbidities, such as preoperative chronic kidney failure and congestive heart failure, should be considered in some cases, as a clear contraindication for compartmental surgery [54,56]. In cancer treatment, patients with a deteriorated PS (≥2) and limited functional capacity have worse tolerance to aggressive cancer treatment, which translates into worse survival outcomes [57]. A study by the European Organization for Research and Treatment of Cancer (EORTC)-Soft Tissue and Bone Sarcoma Group (STBSG) and French Sarcoma Group (FSG) found that PS (≥2) was the most powerful prognostic factor for early mortality in patients with advanced STS treated with chemotherapy [55]. Consequently, for elderly patients with poor PS (≥2) and comorbidities, the indication and extent of surgery must be carefully considered by a multidisciplinary team and the patient himself [56]. As outlined, in pancreatic adenocarcinoma, for elderly patients with locally advanced RPS, the response of the tumor to preoperative treatment, and PS of the patient at the time of surgery, could be used as a measure to select the patients who will benefit the most from surgical salvage [40,58]. Other factors, such as preoperative nutritional status, and the use of early postoperative recovery protocols (ERAS), are currently being integrated into patient selection and optimization criteria prior to surgery [59].

## 6. Morbidity and Complications after Radical Resection of RPS

Several publications have detailed how multivisceral resection in patients with RPS is safe and leads to acceptable short- and long-term morbidity and mortality rates [17,60]. On the other hand, more extensive resection is associated with increased morbidity, especially when three or more organs are resected. In the large TARPSWG series of 1007 consecutive primary RPS resections, overall, 30-day severe morbidity (Clavien–Dindo ≥ 3) and mortality were 16.4% and 1.8%. The most common complications were bleeding, anastomotic leak, and abscess. Significant predictors of severe adverse events were age, need of transfusion, and number of resected organs. Morbidity was correlated with each specific organ, being at highest risk resections involving CPD, major vascular resection, and splenectomy/pancreatectomy [18]. Interestingly, surgical morbidity was not associated with LR/DM or OS. In the recent series of resected RPS published by Callegaro et al., morbidity after RPS resection is described to have decreased in recent years despite the higher number of organs resected, possibly indicating an improvement in perioperative care of RPS patients and better selection of patients for surgery [19].

There are limited data regarding long-term morbidity in patients treated surgically for primary RPS. Multivisceral resections involving the kidney, psoas muscle or nerve, and major vessels could entail long term side effects precluding the quality of life in a disease where more than 60% of patients survive over 5 years. In 2015, Callegaro et al., at the Istituto dei Tumori, Milan, carried out a cross-sectional study to assess long-term morbidity after RPS resection. They assessed the rate of postoperative renal failure, pain, and functional outcomes, reporting that severe chronic pain and lower limb impairment, related to femoral nerve resection, were rare. In multivariate analysis, there was no difference in median levels of creatinine for patients who underwent nephrectomy. Subsequent publications have reported similar outcomes regarding postoperative renal function, demonstrating minimal impact of nephrectomy and adrenalectomy on renal and adrenal function [61,62,63].

## 7. Role of Systemic Treatment and Radiotherapy in RPS

### 7.1. Systemic Therapy

There is no current standard around chemotherapy treatment in RPS, and treatment strategies have been mostly guided by previous experience in STS. However, it is not possible to extrapolate data from STS to RPS due to anatomical constraints and a different disease biology. Neoadjuvant strategy has been suggested for high-risk histologies, such as high-grade dedifferentiated liposarcoma and leiomyosarcoma, as a preventive strategy for hematologic metastasis and, is considered the standard of treatment in some sarcoma centers of reference for borderline resectable high risk RPS [39,64]. The current Strass 2 study is a randomized control trial of neoadjuvant chemotherapy followed by surgery versus surgery alone, to improve disease control and survival in patients with high-risk retroperitoneal sarcoma with an estimated completion date in 2028; nevertheless, previous retrospective reviews have shown conflicting data in this regard [65].

### 7.2. Radiation Therapy

Considering the high risk of microscopic residual disease after surgery, the objective of radiation therapy (RT) in RPS is to decrease risk of local recurrence, which remains the main reason of failure. Policies on timing, dosage, and delivery vary greatly among centers. We lack solid randomized data guiding the use of radiation therapy. The use of external-beam radiation therapy (EBRT) preoperatory is the most accepted approach as it offers several benefits, including more precise target volume, higher biological effect in the absence of altered tissue oxygenation, improved fibrosis of the tumor pseudocapsule easing the detachment of the tumor from critical organs, and the avoidance of treatment delays due to postoperative complications. The results of the multicenter randomized phase 3 STRASS-1 trial were recently published, where neoadjuvant radiation therapy, compared to surgery alone, reported no improvement with the addition of preoperative radiation therapy in abdominal recurrence-free survival (ARFS) compared to surgery alone, for all RPS histologies. However, in the pooled cohort analysis, RT administration was associated with improved ARFS in patients with liposarcoma (*p* < 0.05), specifically, in patients with well-differentiated liposarcoma and G1-2 dedifferentiated liposarcoma. In RPS the combination of EBRT, surgery and intraoperative radiation therapy (IORT) seems to achieve high local control rates, being superior to surgery alone or surgery + EBRT. Preoperative EBRT enhanced with IORT administered to close or positive margins appeared to be the superior strategy in RT for RPS regarding local control and toxicity [50,51,66,67,68].

## 8. Conclusions

Despite great advances in the knowledge of biology and treatment modalities of retroperitoneal sarcoma, surgery remains the standard of care and the only curative treatment for localized disease. Indication and time for surgery in RPS must always be discussed in a multidisciplinary setting, balancing factors such as tumor extension, high risk and aggressive tumoral features, patient age and comorbidities, and the risk of rapid disease progression. Operative procedure aspects are crucial and must be designed based on the patient’s condition, histology, and tumor location in the retroperitoneum. Technically, the surgical approach must be standardized to maximize the chances of obtaining a complete R0 resection with negative microscopic margins [19,22,44].

In the last 15 years, we have observed a significant increase of disease-free survival and OS of patients with resected RPS. The effort and worldwide collaboration of highly specialized sarcoma teams have led to an improved understanding of this disease, as well as to a development of treatment pathways towards personalized treatment. Higher-quality oncological surgery, better patient selection, and perioperative treatment enhancement (i.e., neoadjuvant treatment, intraoperative radiotherapy, and reconstructive surgery) have allowed a paradigm shift on prognosis and quality of life for retroperitoneal sarcoma patients [4,9,29].

## Figures and Tables

**Figure 1 cancers-14-04091-f001:**
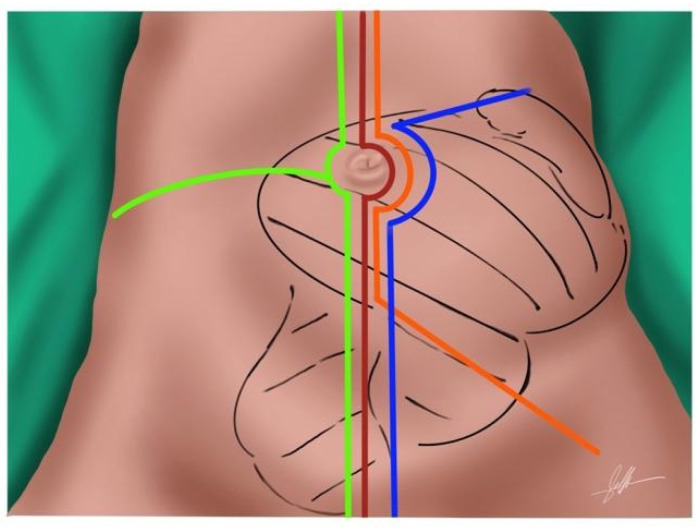
Abdominal incisions for en bloc extended surgery. Midline laparotomy—red—. Flank extension—yellow—. Subcostal extension—blue—. Inferior oblique incision—orange—.

**Figure 2 cancers-14-04091-f002:**
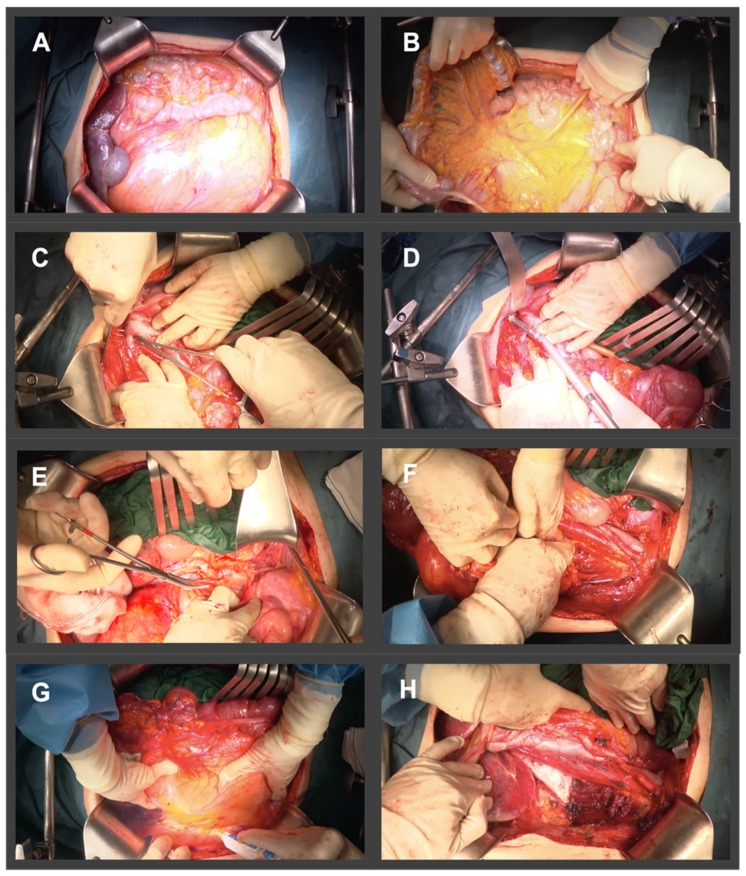
Illustration of surgical steps of an extended en bloc resection for a right Grade 2 DDLPS. (**A**) Extended midline incision. (**B**) Evaluation of a possible intestinal or mesenteric tumor infiltration, with exposure of mesenteric vessels. (**C**,**D**) Dissection and division of renal vessels. (**E**) Dissection and division of distal ureter. (**F**) Dissection of posterior margin with psoas muscle resection preserving femoral nerve. (**G**) Lateral right peritonectomy en bloc with right colon, right kidney en bloc with the tumor. (**H**) Surgical bed.

**Figure 3 cancers-14-04091-f003:**
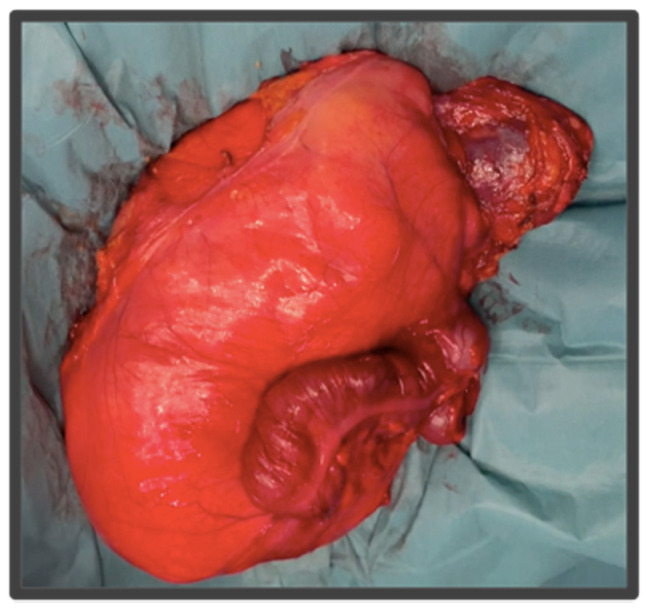
Resected specimen from Figure 2. This right LPS was resected en bloc with the right colon, right kidney, and right psoas muscle.

**Figure 4 cancers-14-04091-f004:**
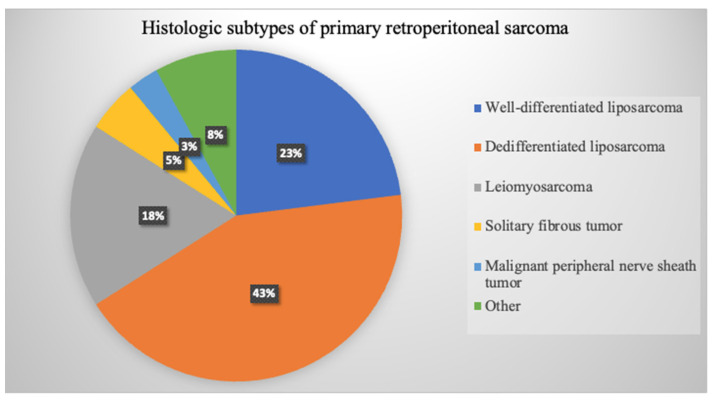
Diagram representation of the most common subtypes of primary retroperitoneal sarcoma based on TARPSWG, 2020 series of 1942 RPS patients [19].

**Table 1 cancers-14-04091-t001:** Adapted from Schmitz, E et al. [44]. Retroperitoneal sarcoma subtypes with their associated pattern of spread, disease failure rate at 5-years, and surgical implications. LR local recurrence, DM distant metastases.

RPS Histology	Pattern of Spread	Disease Failure 5-Year	Surgical Management
WDLPS	Adipose infiltration	LR (19–39%) >> DM (0%)	Extended en-bloc resection requiring ipsilateral retroperitoneal fat resection
	Multilobulated		
	Indistinct borders		
DDLPS	Adipose and visceral infiltration	G2: LR (44%) > DM (10%) G3: LR (33%) << DM (44%)	Extended en-bloc resection requiring ipsilateral retroperitoneal fat resection
	Multilobulated		
	Indistinct borders		
LMS	Distinct borders	LR (6–16%) << DM (55–56%)	En-bloc resection with vascular structures
			May preserve adjacent critical structures
SFT	Distinct borders	LR (4–8%) < DM (17%)	En-bloc resection
			Preservation of adjacent viscera and critical structures
MPNST	Distinct borders	LR (20–35%) > DM (12–13%)	Retroperitoneal approach
			En-bloc resection with associated neurovascular structures
			May preserve adjacent critical structures

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
