# Peer review of "Surgical Principles of Primary Retroperitoneal Sarcoma in the Era of Personalized Treatment: A Review of the Frontline Extended Surgery"

_cancers, 2022, doi:10.3390/cancers14174091_

Round 1

Reviewer 1 Report

Thank you for allowing me to review this manuscript. The authors have satisfactorily addressed all my comments. I have no further revisions. 

Reviewer 2 Report

The authors have revised their original manuscript partly according to the reviewers’ comments. In some points, the authors decided to keep their contents unchanged, however, their rebuttal seems almost reasonable. I would think that this revised manuscript is better organized and suitable for publication.

This manuscript is a resubmission of an earlier submission. The following is a list of the peer review reports and author responses from that submission.

Round 1

Reviewer 1 Report

Thank you for allowing me to review this paper. The authors provide an excellent summary of the recent advances in surgery for retroperitoneal sarcoma. I will start by providing a few general comments and then provide more detailed comments after. In section 3, the authors include very granular details about the surgical technique. I am not sure if this will be of interest to the readership of Cancers. The authors should consider including some discussion around the indications and morbidity of major nerve (e.g., femoral) and vascular reconstruction outside of the context of sarcomas of vascular or nerve origin. It may be beneficial to include a column in Table 1 that describes the indications for resection of these critical structures more commonly associated with morbidity by histologic type. I think the authors should also include a section on the current evidence for perioperative systemic and radiotherapy in the context of primary RPS. The following are my specific comments:

Page 1, abstract: I think the title should be changed to reflect the fact that the manuscript will also discuss the surgical management of recurrent and metastatic disease. I suggest removing the work primary from the title. Otherwise, the authors should consider removing the discussion on recurrent sarcomas. I think this latter suggestion is more appropriate as the surgical management of recurrent or metastatic RPS could be an entire paper on its own.

Page 1, introduction first paragraph: The authors state “Hence, there are no “low risk histologies” as with STS”. I think they are referring to STS in other anatomical locations. This should be clarified.

Page 1, introduction first paragraph: Regarding the discussion around the importance of R0 resections, I would challenge the authors that this has not been as clearly demonstrated in the literature. The authors should include a more balanced discussion around the limitations around pathologic margin assessment and the fact that several historical studies have also found no difference between R0 and R1 margin status.

Page 2, section 2, first paragraph: The authors describe “this extended surgery included the tumor and the adjacent organs located 1-2 cm from this, even if there were not macroscopic organ invasion.” I think they need to specify that it does not involve the removal of organs that would be associated with high morbidity, such as the duodenum or pancreas.

Page 2, section 2, first paragraph: The authors state “Through this approach, the ipsilateral retroperitoneal fat is resected, ensuring the elimination of potential satellite metastases, and optimizing the surgical margin, which also reduces the persistence of microscopic disease and tumor dissemination (3).” The reference cited here does not support this claim. Also, I think that they need to add more uncertainty with regards to the benefits of this frontline approach. There is no randomized controlled data to support that frontline extended surgery achieves superior outcomes. This must be stated and the benefits of frontline extended surgery need to be stated in the context of the available evidence. They should not overstate the known benefits of this approach.

Page 2, section 2, second paragraph: I think there is a typo in “In 2009, these authors presented the first retrospective series from a 20-years period, with the oncological outcomes after the implementation of this new surgical policy on SRP treatment.” I am not sure what SRP means? Its either a typo or an acronym that was not previously defined.

Page 3, section 2, second paragraph: Regarding the discussion of HOI, I think they need to mention the association between HOI and oncologic outcomes and survival to provide a more balanced discussion of the importance of the HOI.

Page 3, section 2: More needs to be said around the current limitations of pathologic assessment of the margins. This is a significant limitation to margin determination and limits the ability to do meaningful studies regarding the benefits of frontline extended surgery.

Page 3, section 3, first paragraph: Is this description based on the consensus from experts? If so, I would make sure the delivery and prose is framed more in line with a group consensus approach to surgery. As it reads now, it feels like this is a description of what is done within a local group or single center. I would include more references throughout this section to help emphasize this is not a single center approach.

Page 4, section 3, first paragraph: I think the authors need to be clearer if they are describing a left or right retroperitoneal sarcoma resection. Presenting the two together is confusing. I see they have the two divided later in the text. I would remove anything from this earlier section that is relevant only for one side versus the other.

Page 4, section 3, paragraph 2: I don’t think everyone routinely resects the entire psoas. Can the authors comment on the benefits of resecting the entire psoas vs. the psoas fascia.

Page 6, section 3, first paragraph: Cattell–Braasch maneuver implies you are preserving the right colon. Can the authors please clarify when they would use a Cattell-Braasch if the right colon is routinely resected for right sided tumors? Do they mean dissection of the small bowel mesenteric attachments to the retroperitoneum?

Page 6, section 4, first paragraph: Are these sentences intended to be included for the readers? It seems like they are more instructions for the authors. Should delete or rephrase.

Page 7, section 4.1, paragraph 3: The authors discuss multifocality. The difference between multifocality and multicentricity in RPS needs to be clearly defined for the readers. Also, the rates of these findings in RPS and the variation by histology need to be clearly described.

Page 7, section 4.1, paragraph 3: The authors comment that atypical lipomatous tumors can undergo marginal resection in the extremities. I do not think that is the case. I think they differ from RPS WDLPS because they can undergo wide local excision with R0 margins decreasing their risk of local recurrence. In the retroperitoneum, WDLPS are difficult to resect with R0 and experts accept that most will be resected with R1. This increases the local failure rate. Thus, both should be resected with the same surgical principles. The difference is that wide local excision is often not feasible in the retroperitoneum. They are the same tumor otherwise.

Page 7, section 4.1, paragraph 4: Regarding the use of perioperative systemic treatment for primary G3 DDLPS, I think the authors need to mention whether or not this is based on randomized controlled data or not. Also, they should specify the evidence to support this and the ongoing STRASS 2 trial that will help answer this question.

Page 7, section 4.1, paragraph 4: The authors state “Of note, in RPS, tumoral grade may be aligned with tumor biology and prognosis more than histology, as in the case of grade 3 DDLPS and grade 3 LMS, sharing similar survival outcomes (35–37).” I think this should be removed or additional evidence should be presented to the authors regarding the plausibility of this.

Page 9, section 4.2, paragraph 1: The authors discuss IORT. I think this requires more details around the feasibility of this and some of the limitations of this method. Any evidence for this also should be included.

Page 9, section 4.2.1, paragraph 1: I don’t think DPC was defined earlier in the paper. Please define.

Page 10, section 5: I would remove the section on management of recurrent disease. This could be a paper in itself.

Page 11, section 6: The authors should include a sentence or two on the importance of international collaboration in advancing the evidence in this rare disease.

Reviewer 2 Report

This is a good review article describing in detail the background of extended surgery for retroperitoneal sarcoma.

Is MPNS at the beginning of the third paragraph on page 8 a misnomer for MPNST? Please confirm.

The authors mention fat resection, but I think that in actual surgery, it is often difficult to distinguish the boundary between normal adipose tissue and neoplastic adipose tissue. Please add how the extent of resection should be considered in light of this background.

I think there is insufficient information on postoperative complications and postoperative quality of life when extended surgery is performed. Please consider adding these information.
